

# Age-associated changes of cytochrome P450 and related phase-2 gene/proteins in livers of rats

Shang-Fu Xu, An-Ling Hu, Lu Xie, Jia-Jia Liu, Qin Wu and Jie Liu

Key Lab for Basic Pharmacology of Ministry of Education, Zunyi Medical University, Zunyi, China

## ABSTRACT

Cytochrome P450s (CYPs) are phase-I metabolic enzymes playing important roles in drug metabolism, dietary chemicals and endogenous molecules. Age is a key factor influencing P450s expression. Thus, age-related changes of CYP 1–4 families and bile acid homeostasis-related CYPs, the corresponding nuclear receptors and a few phase-II genes were examined. Livers from male Sprague-Dawley rats at fetus (−2 d), neonates (1, 7, and 14 d), weanling (21 d), puberty (28 and 35 d), adulthood (60 and 180 d), and aging (540 and 800 d) were collected and subjected to qPCR analysis. Liver proteins from 14, 28, 60, 180, 540 and 800 days of age were also extracted for selected protein analysis by western blot. In general, there were three patterns of their expression: Some of the drug-metabolizing enzymes and related nuclear receptors were low in fetal and neonatal stage, increased with liver maturation and decreased quickly at aging (AhR, Cyp1a1, Cyp2b1, Cyp2b2, Cyp3a1, Cyp3a2, Ugt1a2); the majority of P450s (Cyp1a2, Cyp2c6, Cyp2c11, Cyp2d2, Cyp2e1, CAR, PXR, FXR, Cyp7a1, Cyp7b1. Cyp8b1, Cyp27a1, Ugt1a1, Sult1a1, Sult1a2) maintained relatively high levels throughout the adulthood, and decreased at 800 days of age; and some had an early peak between 7 and 14 days (CAR, PXR, PPARα, Cyp4a1, Ugt1a2). The protein expression of CYP1A2, CYP2B1, CYP2E1, CYP3A1, CYP4A1, and CYP7A1 corresponded the trend of mRNA changes. In summary, this study characterized three expression patterns of 16 CYPs, five nuclear receptors, and four phase-II genes during development and aging in rat liver, adding to our understanding of age-related CYP expression changes and age-related disorders.

## INTRODUCTION

Cytochrome P450s (CYPs) are phase-I metabolic enzymes playing important roles in drug metabolism, dietary chemicals, as well as endogenous molecules in the liver (*Agrawal & Shapiro, 2003*; *Donato & Castell, 2003*; *Hart et al., 2009*; *Munro et al., 2018*). CYP1, 2, 3, and 4 families are responsible for the biotransformation of most foreign substances including 70–80% of all drugs in clinical use, CYP 4 families also participate in lipid metabolism (*Cui, Renaud & Klaassen, 2012*; *Zanger & Schwab, 2013*). The CYP7 families, together with CYP8 and CYP27 are important for cholesterol and bile acid metabolism and homeostasis (*Cuesta de Juan et al., 2007*; *Liu et al., 2014*).

Corresponding author
Jie Liu, Jie@liuonline.com

CYPs are regulated by many physiological, genetic, environmental, and pathological factors. For example, CYPs expression can be affected by hormones (*Daskalopoulos et al., 2012*), cytokines (*Kot & Daujat-Chavanieu, 2018*), pregnancy (*He et al., 2005*), sex (*Agrawal & Shapiro, 2003*; *Das, Banerjee & Shapiro, 2014*) and age. CYPs are subjected to age-dependent changes in cell differentiation (*Czekaj et al., 2010*) and epigenetic regulation (*Li et al., 2009*), and age-related metabolic syndrome (*Bondarenko et al., 2016*), kidney diseases (*Velenosi et al., 2012*), diabetes (*Park et al., 2016*), nonalcoholic steatohepatitis (*Li et al., 2017*), virus hepatitis, and cirrhosis (*Kirby et al., 1996*).

Age of animals greatly affects drug metabolism (*Durnas, Loi & Cusack, 1990*), alters pharmacokinetics of xenobiotics (*Matalova, Urbanek & Anzenbacher, 2016*; *Shi & Klotz, 2011*), and thus alters the sensitivity to drugs and toxicants such as acetaminophen (*Mach et al., 2014*), isoniazid (*Mach et al., 2016*), aflatoxin B1 (*Kirby et al., 1996*; *Wang et al., 2018*), and thioacetamide (*Kang et al., 2008*). Age also influences drug-drug interactions (*Jia et al., 2014*). Age-associated changes in P450 and corresponding nuclear factors are a major determinant in CYP regulation of drug metabolism, especially during development (children) and in senescence (elderly) (*Durnas, Loi & Cusack, 1990*; *Kilanowicz et al., 2015*; *Shi & Klotz, 2011*).

The expression and maturation of CYPs during development is a major topic of research (*Kilanowicz et al., 2015*), and immature rats have been proposed as a potential model for xenobiotics risk evaluation for children (*McPhail et al., 2016*). The ontogeny of CYPs greatly affects the drug metabolism especially during the developmental period (*De Zwart et al., 2008*), and is the major cause of altered susceptibility to drugs and toxicants in children (*Li et al., 2017*; *Yun et al., 2010*).

Aging is a physiological process characterized by a progressive functional decline in various organs over time. Aging is an important factor leading to alterations in the biotransformation, either by reduced expression or decreased function. Many cytochrome P450 genes from CYP 1–3 families show decreased expression in the older rats.(*Yun et al., 2010*) The ability of liver CYPs to metabolize xenobiotics decreases with aging *in vitro* (*Salmin et al., 2017*) and *in vivo* (*Wauthier, Verbeeck & Calderon, 2007*), and hepatic CYP mRNA expressions are decreased with aging (*Mori et al., 2007*). The ability of CYPs in response to inducers such as phenobarbital is also decreased in old rats (*Agrawal & Shapiro, 2003*). Age-associated CYP3A expression changes in the liver are more remarkable as compared to that occurred in the intestine and kidney, and are tissue-specific (*Warrington, Greenblatt & Von Moltke, 2004*). Since P450 enzymes in humans are regulated in a manner similar to that in animals (*Durnas, Loi & Cusack, 1990*; *Wauthier, Verbeeck & Calderon, 2007*). Thus, to examine CYP expressions in the old laboratory animals would help evaluation of drug metabolism, efficacy and toxicity in the elderly.

In children, three patterns of drug metabolizing enzymes are proposed (*Hines, 2008*). The first pattern (e.g., CYP3A7) is expressed at the highest level during the first trimester and either remains at high concentrations or decreases during gestation, but is silenced or reduced within one to two years after birth; the 2nd pattern (e.g., SULT1A1) is expressed at relatively constant levels throughout gestation and minimal changes are observed postnatally; and the 3rd pattern (e.g., ADH1C) is not expressed or is expressed

at low levels in the fetus (*Hines, 2008*; *Hines, 2013*). Age-associated changes of drug metabolism in humans, especially during stages before birth and during early development (neonate/infant/child), could be studied in laboratory animals (*Hines, 2013*; *Saghir, Khan & McCoy, 2012*).

Considering the variations of P450 in children, adult, and elderly, this study was initiated to characterize age-associated changes in hepatic P450, to extend our prior work on age-associated changes in hepatic uptake Oatp transporters (*Hou et al., 2014a*; *Hou et al., 2014b*), efflux MRP transporters. *Zhu et al. (2017)*, the Nrf2 antioxidant pathways (*Xu et al., 2018b*), glutathione S-transferases (*Xu et al., 2018a*), and the antioxidant metallothionein gene expression (*Hou et al., 2014a*; *Hou et al., 2014b*). The expression of 16 major CYP isoforms, five corresponding nuclear receptors (NRs), and four phase-II conjugation genes were examined. Protein expressions of selected CYPs were also performed to confirm qPCR results. Similar to three patterns of CYP expression during mouse liver development (*Hart et al., 2009*), the current study identified three patterns of CYP expression in the liver of rats at 11 time points of entire life span to help our understanding the age-associated changes in these important phase-I and phase-II drug metabolism genes, and age-associated disorders.

## MATERIALS AND METHODS

### Animals
Adult male and female SD rats (250–300 g, 10 males and 30 females) were purchased from the Experimental Animal Center of Third Military Medical University (Chongqing, China; Certificate No: CXK 2007–0005). Rats were kept in a SPF-grade animal Facilities with controlled environment (22 $\pm$ 1 °C, 50 $\pm$ 2% humidity and a 12 h: 12 h light: dark cycle) at Key Lab for Basic Pharmacology of Ministry of Education. Rats had free access to purified water and standard laboratory chow (Experimental Animal Center, Chongqing, China). All animal care and experimental protocols were complied with the Animal Management Guidelines of China and approved by the Animal Use and Care Committee of Zunyi Medical University (2012-02).

### Sample collection
Rats were acclimatized for one week before timely mating overnight and a positive vaginal plug next morning was considered as gestation day 1. Livers of offspring male rats were collected at gestation day 19 (−2 d), at birth (1 d), at the neonatal stage (7 and 14 d), at weanling (21 d), at puberty (28 and 35 d), at the adulthood (60 and 180 d), and at aging (540 and 800 d). Six samples per time point were collected, however, $n = 4$–5 was used to for a 96-well qPCR plate to hold all time points. Rats were anesthetized by hloral hydrate (10%, 5 mL/kg, ip), followed by decapitation to minimize potential pain and distress. Liver tissues were stored at −80 °C prior to analysis.

### Real-time RT-PCR analysis
Liver total RNA was extracted by using RNAiso Plus kit (TaKaRa Biotechnology Co., Ltd., Dalian, China). The quality and quantity of total RNA were determined by nanodrop

and the 260/280 nm ratio >1.8. The total RNA was reverse transcribed to cDNA (Applied Biosystems, Foster City, CA, USA) and real-time RT-PCR analysis (Bio-Rad Laboratories, Hercules, CA, USA) was conducted as described (*Xu et al., 2018b*). Relative expression of genes was calculated by the $2^{-\Delta\Delta Ct}$ method and normalized to the house-keeping gene β-actin or GAPDH (results were similar, data not shown), and expressed as relative transcript levels, setting controls as 100%. The primer sequences used in this study were shown in the Table S1 including CYP1 family gene Cyp1a1, Cyp1a2 and corresponding nuclear receptor AhR; CYP2 family gene Cyp2b1, Cyp2b2, Cyp2c6, Cyp2c11, Cyp2d2, Cyp2e1 and corresponding nuclear receptor CAR; CYP3 family genes Cyp3a1,Cyp3a2 and corresponding nuclear receptor PXR; CYP4 family Cyp4a1 and nuclear receptor PPARα; as well as CYPs for bile acid metabolism Cyp7a1, Cyp7b1, Cyp8b1,Cyp27a1 and corresponding nuclear receptor FXR. In addition, the phase 2 genes for glucuronidation Ugt1a1 and Ugt1a2, sulfation (Sult1a1 and Sult1a2) were also examined.

## Western blot analysis

Liver tissues (50–100 mg) were homogenized in RIPA lysis buffer (Beyotime Institute of Biotechnology, Shanghai, China) containing 1 mM phenylmethanesulfonyl fluoride (PMSF) and freshly prepared proteinase inhibitors. Protein concentrations were quantified by the BCA assay (Beyotime Institute of Biotechnology, Shanghai, China) and denatured (90 °C, 10 min with Nupage Loading buffer). Aliquoted proteins (30 μg) were separated on NUPAGE 10% BT gels (Thermo Fisher Scientific, Waltham, MA, USA) and transferred to PVDF membranes. After blocking with 5% nonfat milk at room temperature for two hours, membranes were incubated with primary mouse antibody against β-actin, rabbit polyclonal antibodies against rat CYP1A2 (bs-2589R), CYP2B1 (bs-14177R), CYP2E1 (bs-4562R), CYP3A1 (bs-20586R), CYP4A1 (bs-5054R) and CYP7A1 (bs-21429R) (1:1000) (Biosynthesis Biotechnology Co., LTD. Beijing, China) overnight at 4 °C. After washes with TBST, membranes were incubated with horseradish peroxidase conjugated anti-rabbit, anti-mouse IgG secondary antibodies (1:5000) for 1 h at room temperature. Protein antibody complexes were visualized using an Enhanced Chemiluminescent reagent and a ChemiDoc XRS system (Bio Rad Laboratories, Inc., Hercules, CA, USA). Band intensities were semi-quantified by densitometry using Quantity One® software (version 4.6.2; Bio Rad Laboratories, Inc., Hercules, CA, USA) (*Xu et al., 2018a*).

## Statistical analysis

The software SPSS version 16.0 (SPSS, Inc., Chicago, IL, USA) was used for statistical analysis. Data were expressed as the mean $\pm$ SEM ($n = 4$–5 per time point). Age associated differences were analyzed by one-way analysis of variance, followed by the least significant difference post hoc test, $p < 0.05$ was considered to indicate a statistically significant difference from the levels of birth.

# RESULTS

## Age-related expression of CYP-1 family

The expression of CYP-1 family (Cyp1a1, Cyp1a2, and corresponding nuclear receptor AhR) is shown in Fig. 1. Aryl hydrocarbon receptor (AhR) mainly mediates the expression

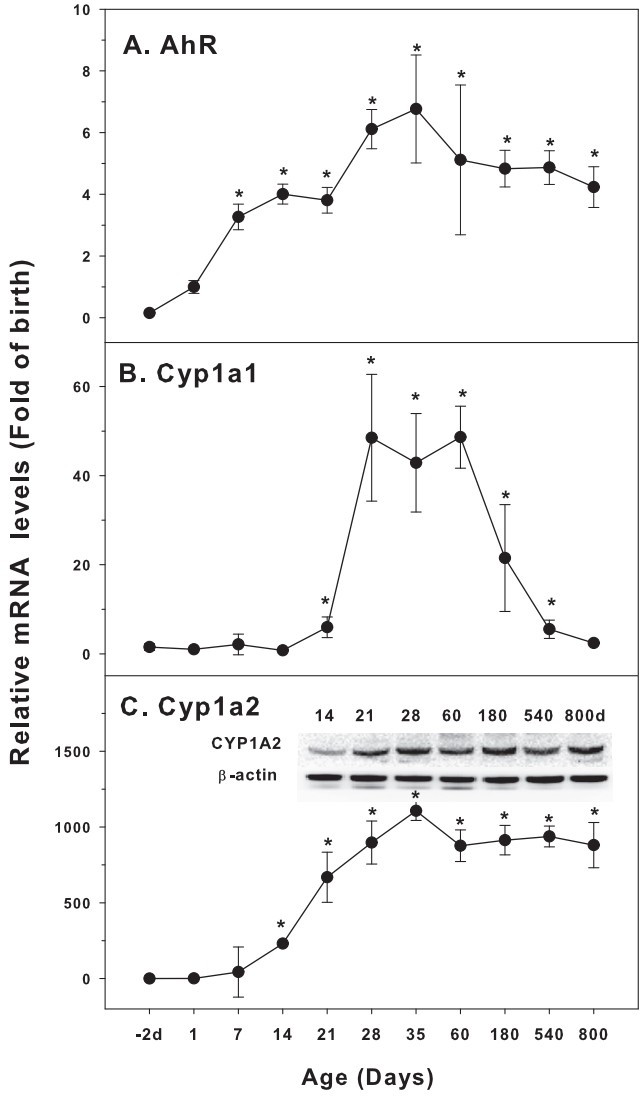

**Figure 1** **Age-related expression of CYP-1 family gene/proteins in livers of male rats.** (A) AhR, (B) Cy1a1, (C) Cyp1a2. Livers from male SD rats at the fetus ($-2$ d before birth), the neonatal stage (1, 7, and 14 d), and at weanling (21 d), at puberty (28 and 35 d), at adult (60 and 180 d), and at aging (540 and 800 d), were collected to extract RNA, followed by qPCR analysis ($n = 4$–$5$ for each time point). *Significantly different from at birth, $p < 0.05$. For western blot insert, the neonatal (14 d), at weanling (21 d), at puberty (28 d), at adult (60 and 180 d), and at aging (540 and 800 d) were collected to extract protein. Aliquoted proteins (30 μg) were separated on NUPAGE 10% BT gels and the representative western blot was inserted into the figure ($n = 3$). The molecular weight for CYP1A1 was 55 kD, and β-actin 43 kD.

Full-size [image] DOI: 10.7717/peerj.7429/fig-1

of CYP1A1 and CYP1A2 proteins. 2,3,7,8-Tetrachlorodibenzodioxin (TCDD) is a typical CYP1A1 inducer. When TCDD is combined with AhR, AhR is dissociated from the complex and transferred to the nucleus. It forms heteromeric dimers with AhR nuclear transport protein, and then induces the expression of target genes. In this way, TCDD and other AhR activators significantly induce the expression of CYP1A genes (*Aleksunes & Klaassen, 2012*). AhR was low in fetal livers ($-2$ d), and gradually increase after birth, and reached the

highest levels at 35 days of age (6.7-fold of birth), and gradually declined, and at 800 days of age, it remained 4.2-fold higher than at birth (Fig. 1A). Cyp1a1 was low at −2 days of age through 14 days of age, and begin to increase at 21 days of age, rapidly increased 48-fold at 28–60 days of age, and decline rapidly after 60 days of age, and returned to 2.4-fold of the birth level at 800 days of age (Fig. 1B). In contrast, Cyp1a2 followed the similar pattern as AhR. Cyp1a2 increased dramatically after birth, reached 250-fold at weanling (21 day), and peaked on 35 days of age (1,100 fold). It was declined to 850 fold of birth at 60 days of age and remained till 800 days of age. CYP1A2 protein expression followed the similar pattern (Fig. 1C).

## Age-related expression of CYP-2 family

The expression of CYP-2 family is shown in Fig. 2. Constitutive androstane receptor (CAR) is a nuclear receptor of steroid hormones. It regulates the metabolizing enzymes and transporters in liver and small intestine. CAR mediates endogenous hormone or exogenous drug reactions, such as phenobarbital, and transcriptionally regulates CYP2 expression (*Aleksunes & Klaassen, 2012*). CAR was low in fetal livers (−2 d), and gradually increase after birth, first peaked at 7 days of age, and reached the highest levels at 60 days of age (6-fold of birth), and gradually declined, and at 800 days of age, it remained 2.3-fold higher than at birth (Fig. 2A). Cyp2b2 was low in fetal livers (−2 d), and begin to increase after birth, and rapidly increased at weaning (21 d), reaching the peak at 35 days of age (7-fold of birth) and decreased afterwards, and returned to birth level after 540 days of age (Fig. 2B). The expression of Cyp2b1 followed the similar pattern as Cyp2b2, and the expression of CYP2B1 protein followed the similar pattern (Fig. 2C). Cyp2c6 increased after weanling, reached the peak of liver maturation (350 fold) at 60 days of age, and remained high till 540 days of age, and decreased to 250-fold over birth levels at 800 days of age (Fig. 2D). Cyp2c11 increased 800-fold at puberty (35 days of age), but dramatically increased with liver maturation (60,000-fold at 60 days, 98,000-fold at 180 days, and 110,000-fold at 540 days of age), and rapidly decreased at aging of 800 days of age, but it was still 40,000-fold over the birth level. (Fig. 2E). Cyp2d2 increased gradually after birth, reached the peak (9-fold) of liver maturation at 35–180 days of age, and decreased to 6-fold of birth at 540 and 800 days of age (Fig. 2F). The expression pattern of Cyp2e1 was relatively stable: Cyp2e1 increased rapidly after birth, reached 30-fold of the birth levels at weanling (21 days of age), and peaked on 35 days of age (35 fold). It was gradually declined but remained at the high level at the age of 800 days (22 fold of the birth level). CYP2E1 protein expression followed the similar pattern (Fig. 2G).

## Age-related mRNA expression of CYP-3 family

The expression of CYP-3 family is shown in Fig. 3. Pregnane X receptor (PXR) is a highly conserved ligand-dependent transcription factor. It is mainly expressed in the liver and partly expressed in the colon and small intestine. The regulation of CYP3A by PXR-mediated signaling pathway is an important pathway of drug metabolism (*Aleksunes & Klaassen, 2012*). The expression of PXR was relatively stable throughout the life with approximately 2-fold variations (Fig. 3A). The expression of Cyp3a2 markedly increased

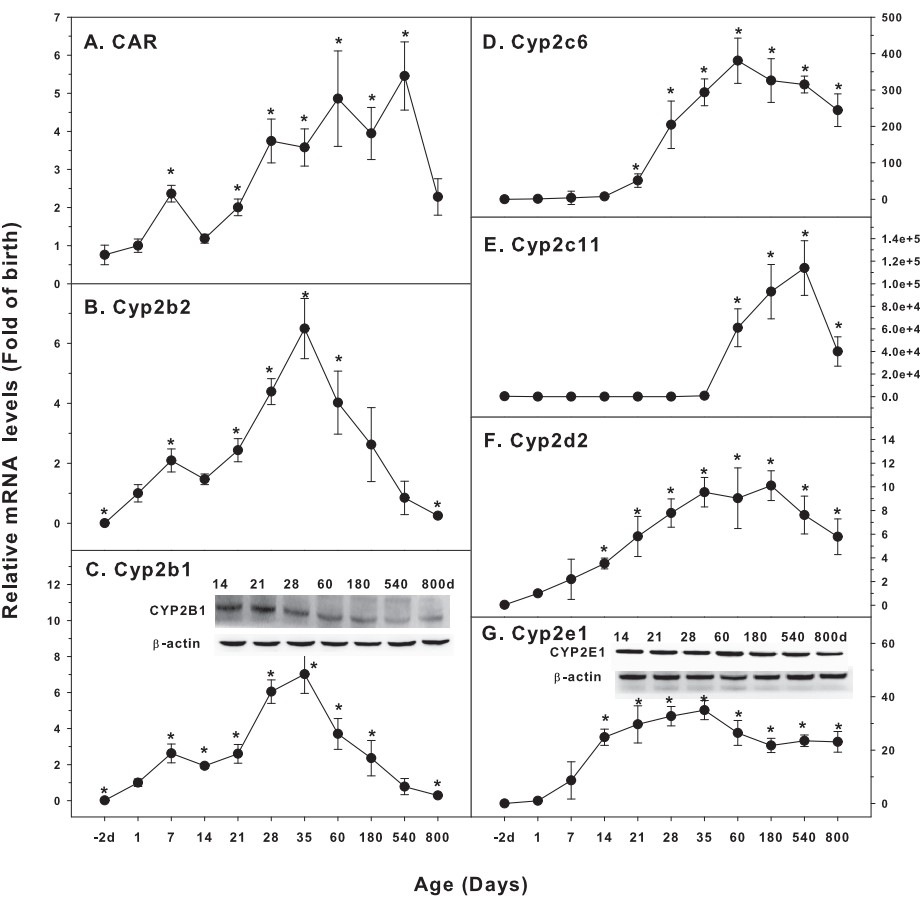

**Figure 2** **Age-related expression of CYP-2 family gene/proteins in livers of male rats.** (A) CAR, (B) Cyp2b2, (C) Cyp2b1, (D) Cyp2c6, (E) Cyp2c11, (F) Cyp2d2, (G) Cyp2e1. Livers from male SD rats at the fetus ($-2$ d before birth), the neonatal stage (1, 7, and 14 d), and at weanling (21 d), at puberty (28 and 35 d), at adult (60 and 180 d), and at aging (540 and 800 d), were collected to extract RNA, followed by qPCR analysis ($n = 4$–5 for each time point). *Significantly different from at birth, $p < 0.05$. For western blot insert, the neonatal (14 d), at weanling (21 d), at puberty (28 d), at adult (60 and 180 d), and at aging (540 and 800 d) were collected to extract protein. Aliquoted proteins (30 μg) were separated on NUPAGE 10% BT gels and the representative western blot was inserted into the figure ($n = 3$). The molecular weight for CYP2B1 was 56 kD, CYP2E1 57 kD, and β-actin 43 kD.

4 fold at 7 days of age, and rapidly increased after weanling (21 days of age), reached the peak at 28 days of age (25-fold of birth) and decreased gradually afterwards, and the level was still 5.4-fold of the birth level at 800 days of age (Fig. 3B). Cyp3a1 follows similar pattern as Cyp3a2. It was markedly increased after weanling, reached the peak at 28 days of age (13-fold of the birth level) and decreased gradually afterwards, and the level was still 3.4-fold of the birth level at 800 days of age. The expression of Cyp3a1 protein followed the similar patter (Fig. 3C).

## Age-related expression of CYP-4 family

The expression of CYP-4 family is shown in Fig. 4. Peroxisome proliferator-activated receptors (PPARs) nuclear receptor family regulates the expression of genes that control

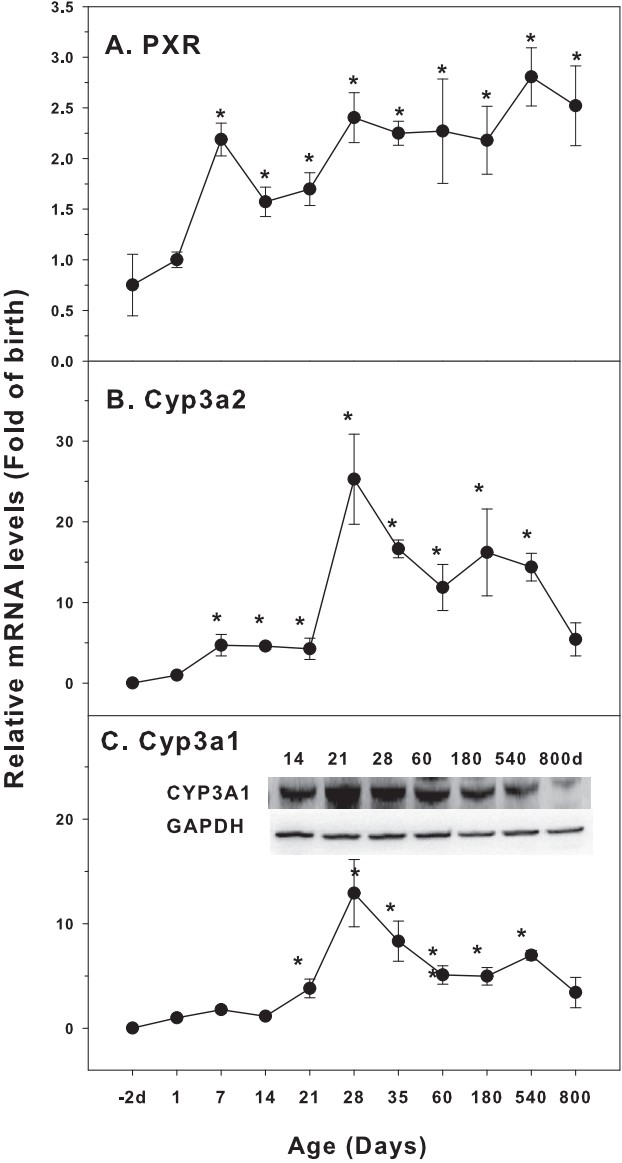

**Figure 3** **Age-related expression of CYP-3 family gene/proteins in livers of male rats.** (A) PXR, (B) Cyp3a2, (C) Cyp3a1. Livers from male SD rats at the fetus (−2 d before birth), the neonatal stage (1, 7, and 14 d), and at weanling (21 d), at puberty (28 and 35 d), at adult (60 and 180 d), and at aging (540 and 800 d), were collected to extract RNA, followed by qPCR analysis ($n = 4$–5 for each time point). *Significantly different from at birth, $p < 0.05$. For western blot insert, the neonatal (14 d), at weanling (21 d), at puberty (28 d), at adult (60 and 180 d), and at aging (540 and 800 d) were collected to extract protein. Aliquoted proteins (30 μg) were separated on NUPAGE 10% BT gels and the representative western blot was inserted into the figure ($n = 3$). The molecular weight for CYP3A1 was 57 kD, and β-actin 43 kD.

fatty acid synthesis, storage, and catabolism. PPARs mainly include PPARα, PPARβ and PPARγ. The activation of PPAR can improve insulin resistance, slow down atherosclerosis, and promote the metabolism of cholesterol. PPARα regulates induction of CYP4A gene (*Aleksunes & Klaassen, 2012*). The expression of PPARα was relatively stable throughout

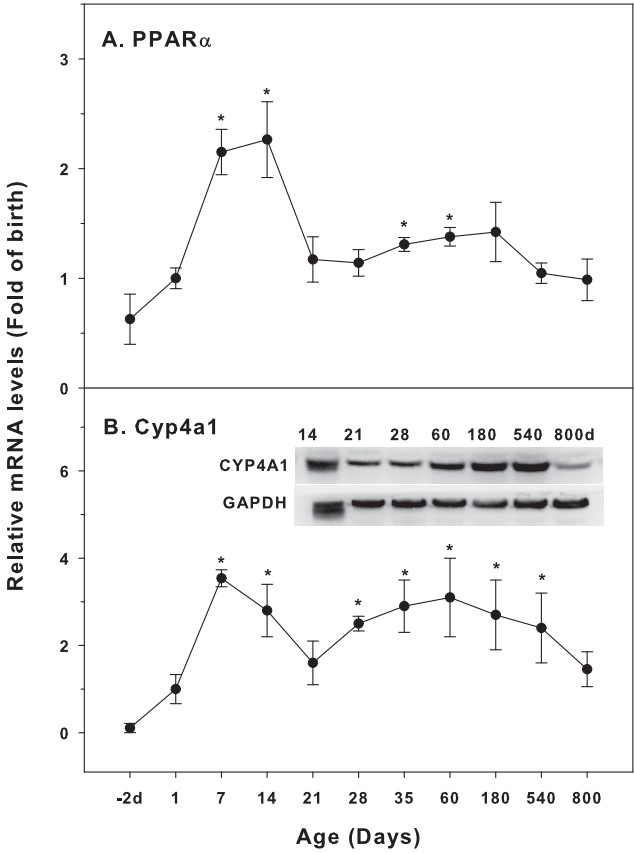

**Figure 4   Age-related expression of CYP-4 family gene/proteins in livers of male rats.** (A) PPARα, (B) Cyp4a1. Livers from male SD rats at the fetus (−2 d before birth), the neonatal stage (1, 7, and 14 d), and at weanling (21 d), at puberty (28 and 35 d), at adult (60 and 180 d), and at aging (540 and 800 d), were collected to extract RNA, followed by qPCR analysis ($n = 4$–5 for each time point). For western blot insert, the neonatal (14 d), at weanling (21 d), at puberty (28 d), at adult (60 and 180 d), and at aging (540 and 800 d) were collected to extract protein. Aliquoted proteins (30 μg) were separated on NUPAGE 10% BT gels and the representative western blot was inserted into the figure ($n = 3$). The molecular weight for CYP4A1 was 59 kD, and β-actin 43 kD.

the life, except for 7 and 14 days of age (2-fold of the birth level), and at 800 days of age, its levels returned to the birth level (Fig. 4A). Cyp4a1 started to increase after birth, with the first peak at 7 days (3.5-fold) and decreased after 14 days of age, and again gradually increased after weanling and remained high throughout 540 days of age. At 800 days of age, it was still 1.8-fold of the birth level. The expression of CYP4A1 protein followed the similar pattern (Fig. 4B).

## Age-related mRNA expression of CYPs involved in bile acids homeostasis

The expression of CYPs involved in cholesterol and bile acids homeostasis is shown in Fig. 5. Bile acids (BAs) are the endogenous ligands of farnesoid X receptor (FXR), so FXR is also called the BA receptor. Cholesterol 7α hydroxylase (Cyp7a1) is important for BA

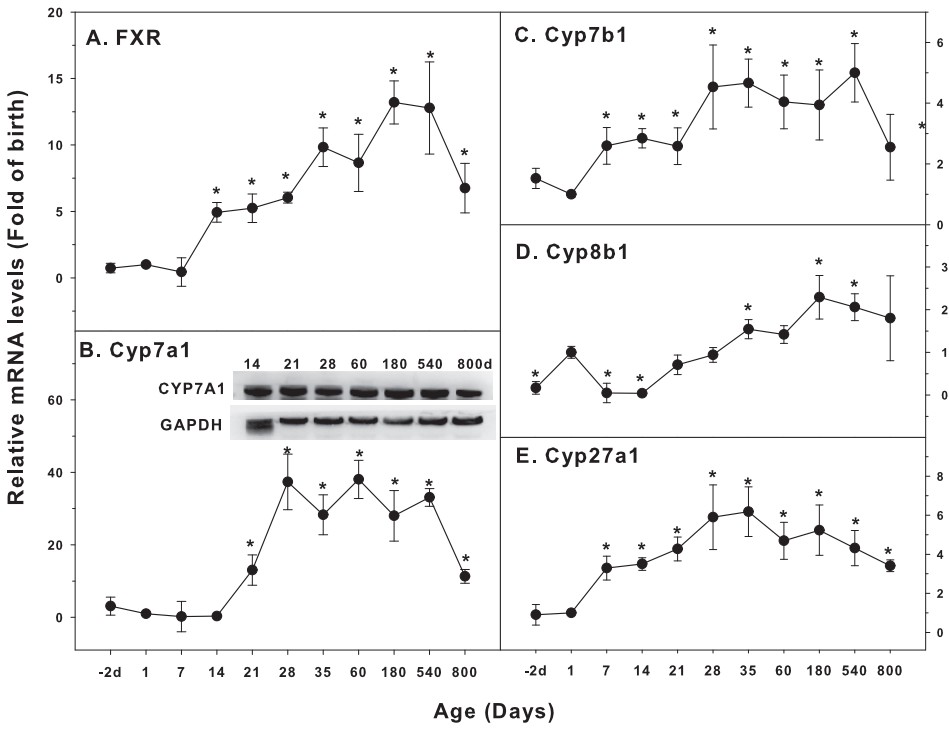

**Figure 5  Age-related expression of CYP-7 family gene/proteins in livers of male rats.** (A) FXR, (B) Cyp7a1, (C) Cyp7b1, (D) Cyp8b1, (E) Cyp27a1. Livers from male SD rats at the fetus ($-2$ d before birth), the neonatal stage (1, 7, and 14 d), and at weanling (21 d), at puberty (28 and 35 d), at adult (60 and 180 d), and at aging (540 and 800 d), were collected to extract RNA, followed by qPCR analysis ($n = 4$–5 for each time point). *Significantly different from at birth, $p < 0.05$. For western blot insert, the neonatal (14 d), at weanling (21 d), at puberty (28 d), at adult (60 and 180 d), and at aging (540 and 800 d) were collected to extract protein. Aliquoted proteins (30 μg) were separated on NUPAGE 10% BT gels and the representative western blot was inserted into the figure ($n = 3$). The molecular weight for CYP7A1 was 55 kD, and β-actin 43 kD.

synthesis. When BA overloads in the liver, toxicity to liver cells occurs, including oxidative stress, inflammation, necrosis and even cirrhosis (*Cuesta de Juan et al., 2007*; *Liu et al., 2014*). The expression of FXR increased at 14 days of age (5 fold), and gradually increased afterwards with age, reached peak at 180 days (13 fold), and still high at 800 days (6.8 fold) (Fig. 5A). Cyp7a1 started to increase at 14 days of age, reached the peak at 28 days (37 fold of birth), and remained high throughout the adulthood, and was still 11-fold higher than the birth level at 800 days of age. The expression of CYP7A1 protein followed similar pattern (Fig. 5B). The expression of Cyp7b1 started to increase after birth, reached the peak at 28 days of age, and decreased at 800 days of age (Fig. 5C). Cyp8b1 increased at birth and decreased at 7 days of age. Cyp8b1 started to increase again after weanling, reached the peak at adulthoods at 180 days of age, and gradually decreased thereafter. It was still higher at 800 days of age (Fig. 5D). The expression of Cyp27a1 held a similar pattern, reached the peak on 35 days of age (5.8-fold over the birth level), and remained the high levels

throughout 540 days of age, and decreased at 800 days of age with 3-fold higher over the birth level (Fig. 5E).

## Age-related mRNA expression of UGT and SULT families

The expression of UGT and SULT families is shown in Fig. 6. UDP-glucuronosyltransferases (UGTs) and sulfotransferases (SULTs) are the two most important phase-2 conjugation enzymes to conjugate the CYP catalyzed metabolites and drugs contain functional groups such as hydroxyls and carboxylic acids (*Coughtrie, 2015*). Glucuronidation involves the reaction of uridine 5′-diphosphoglucuronic acid with a number of functional groups generated from CYP metabolism and is a major mechanism for the formation of water-soluble substrates for their elimination in bile or in urine, especially for the clearance of a number of drugs in children (*Krekels et al., 2012*). SULTs transfer the sulfuryl moiety from the universal donor PAPS (3′-phosphoadenosine 5′-phosphosulfate) to a wide variety of substrates with hydroxyl- or amino-groups after CYP metabolism (*Coughtrie, 2016*). Ugt1a1 was low in fetal livers (−2 d), and gradually increase after birth, marked increase after weaning (21 days), and reached the highest levels at 28 days of age (7.8-fold of birth), and gradually declined, and at 800 days of age, it remained 4-fold higher than the level of birth (Fig. 6A). The expression of Ugt1a2 was relatively stable throughout the life, except for a small peak at day 28 and 35 days of age (2.5-fold higher than the birth level) (Fig. 6B). The expression of Sult1a1 started to increase at 14 days of age, reached the peak at 180 days of age (20-fold), remained 14-fold higher than the birth levels (Fig. 6C). The expression of Sult1a2 started to increase at 14 days of age, reached the first peak at weaning (15 fold), but dramatically increased after 35 days of age, reached 110-fold at 60 days of age, and 190-fold at 540 days of age. It remained 70-fold higher than the levels of birth (Fig. 6D). The expression pattern of Sult1a1 was quite similar to the expression pattern of Cyp2c11.

## DISCUSSION

The present study characterized age-related expression of 16 CYPs, 5 NRs, and 4 phase-II genes in livers of rats across 11 time points from fetus (−2 d), neonates (1, 7, 14 and 21 d), puberty (28 and 35 d), adulthood (60 and 180 d), to aging (540 and 800 d). In general, there are three patterns of their expression: (1) The expressions of AhR, Cyp1a1, Cyp2b1, Cyp2b2, Cyp3a1, Cyp3a2, and Ugt1a2 were low in fetal and neonatal stage, increased with liver maturation and decreased extensively after 180 days; (2) the majority of CYPs and other genes maintained relatively high levels throughout the adulthood and decreased at aging of 540 and/or 800 days; (3) the expression of CAR PXR, PPARα, Cyp4a1, Ugt1a2 had a first peak between 7–14 days of age. The protein expression of CYP1A2, CYP2B1, CYP2E1, CYP3A1, CYP4A1, and CYP7A1 followed the trend of mRNA changes. Characterization of CYPs in rat entire life span provides fundamental information for drug metabolism and pharmacology studies in children and elderly.

There are three unique features of the current study: (1) Most of the studies on age-related changes in hepatic CYPs are performed in mice (*Cui, Renaud & Klaassen, 2012*; *Hart et al., 2009*; *Li et al., 2009*), this study characterized CYPs and NRs in rats, another commonly-used laboratory animals; (2) Most of the age-related changes in hepatic P450

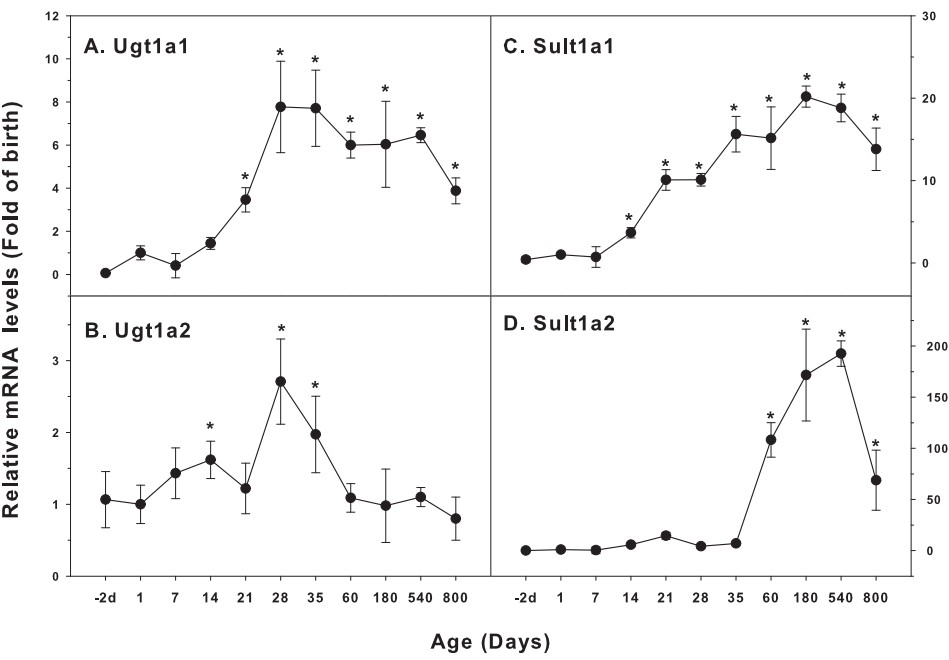

**Figure 6  Age-related mRNA expression of UGT and SULT family genes in livers of male rats.** (A) Ugt1a1, (B) Ugt1a2, (C) Sult1a1, (D) Sult1a2. Livers from male SD rats at the fetus (−2 d before birth), the neonatal stage (1, 7, and 14 d), and at weanling (21 d), at puberty (28 and 35 d), at adult (60 and 180 d), and at aging (540 and 800 d), were collected to extract RNA, followed by real-time RT-PCR analysis ($n = 4$–5 for each time point). *Significantly different from at birth, $p < 0.05$.

are focused on the developmental stages to maturation (*Asaoka et al., 2010*; *Bondarenko et al., 2016*; *Cui, Renaud & Klaassen, 2012*; *De Zwart et al., 2008*; *Kilanowicz et al., 2015*; *Park et al., 2016*), and this study covered the whole life span, and (3) this study extended our efforts to characterize age- and sex-related changes in hepatic drug transporters (*Hou et al., 2014a*; *Hou et al., 2014b*; *Zhu et al., 2017*) and defense mechanisms (*Hou et al., 2014a*; *Hou et al., 2014b*; *Xu et al., 2018a*; *Xu et al., 2018b*).

## P450-1 family

CYP1 family is responsible for activation of toxicants and drugs. Cyp1a1 is expressed very early in rodents and involved in developmental toxicity of hexachloronaphthalene (*Kilanowicz et al., 2015*), and immature rats has been proposed as a potential model for chemical risks in children (*McPhail et al., 2016*). Ontogeny of hepatic CYP1A2 showed it rapidly increased after weaning, followed by significant decrease during adulthood (*Elbarbry, McNamara & Alcorn, 2007*), However, CYP1A1/2 activity did not change when fed Zuker diabetic fatty rats with high fat diet at 5 week (insulin resistant stage) and 11-week (diabetic stage) (*Park et al., 2016*). Glycyrrhetinic acid potentiation of clozapine hepatotoxicity is associated with CYP1A2 induction and suppression of CYP2C11 and 2C13 (*Jia et al., 2014*). CYP1A1 and CYP1A2 are regulated differently, CYP1A1 decreased rapidly after maturation, while CYP1A2 remained the high expression levels till 104 weeks (*Yun et al., 2010*). The present results agreed with the literature.

## P450-2 family

Hepatic CYP2C2 and CYP2C11 decreased with age, along with CYP3A enzyme genes (*Mori et al., 2007*). In diethyl nitrosamine-induced liver insufficiency, the altered expression of cytokines might contribute to CYP2C and CYP3A isoform regulation (*Kot & Daujat-Chavanieu, 2018*). The expression of CYP2C11 is under the regulation of growth hormones (*Das, Banerjee & Shapiro, 2014*), as well as the dopaminergic receptors (*Daskalopoulos et al., 2012*). The expressions of CYP2C, CYP2E1, and CYP3A are influenced by metabolic syndrome (*Bondarenko et al., 2016*). CYP2C11 is a male-specific CYP, its expression began to increase after weanling, and at puberty reached 830-fold over the birth, and further dramatically increased over 100,000-fold over the birth level throughout the adulthood, but rapidly decreased at aging, consistent with the literature (*Agrawal & Shapiro, 2003*; *Das, Banerjee & Shapiro, 2014*; *Yun et al., 2010*).

Age-associated CYP2E1 expression has important implications in pharmacology and toxicology. Old rats have decreased CYP2E1, altered acetaminophen pharmacokinetics, resulting in less sensitive to acetaminophen toxicity (*Mach et al., 2014*). On the other hand, higher CYP2E1 in young rats might be a reason of increased sensitivity to isoniazid toxicity (*Mach et al., 2016*). The observed expression pattern for CYP2B1 (marked decreases at 800 days of age) and CYP2E1 (slight decreases at 800 days of age) are in agreement of the literature (*Yun et al., 2010*). CYP2E1 plays roles in thioacetamide hepatotoxicity, as Cyp2e1-/-mice are less sensitive to liver injury (*Kang et al., 2008*). Age-related differences in CYP2E1 levels can have important implications for the toxic and carcinogenic actions of some hydrocarbons (e.g., benzene, hexane) and short-chain halocarbons, such as carbon tetrachloride (*McPhail et al., 2016*).

## P450-3 family

CYP3 families are responsible for most drug metabolism (*Zanger & Schwab, 2013*). In rats with chronic kidney diseases, CYP3A and CYP2C mediated metabolism are decreased (*Velenosi et al., 2012*). Using liver microsomes (S9) from young and old rats, severe metabolism impairment with aging for CYP3A and CYP2D substrates are observed (*Salmin et al., 2017*). CYP3A1 is higher in cells from young rats than in old rats (*Czekaj et al., 2010*). CYP3A in the liver are sensitive to aging with 50–70% decreases, while CYP3A in the intestine is unchanged and in the kidney increased (*Warrington, Greenblatt & Von Moltke, 2004*). Generally speaking, hepatic CYP3A enzymes are decreased with age (*Mori et al., 2007*). In the present study, age-associated changes of CYP3A1 and CYP3A2 followed "Pattern 1", while the expression of PXR, Cyp3a11 (mouse), and CYP3A4 protein followed "Pattern 2", a phenomenon in agreement of the literature (*Mori et al., 2007*; *Yun et al., 2010*).

## P450-4 family

PPARα activation induces CYP4A1, together with acyl-CoA oxidase and SREBPs, that play important roles in regulating lipid metabolism, especially in rats fed with high-fat-diet (*Chang et al., 2011*). The herbicide propaquizafop dose-dependently activates PPARα and CYP4A, leading to increased liver weight and hypertrophy as a mode of action in

hepato-carcinogenesis (*Strupp et al., 2018*). In addition to PPARα activation, the induction of CYP4A by K$^+$PFOS also involves CAR and PXR activation (*Elcombe et al., 2012*), leading to hepatomegaly. In the present study, the expression of PPARα, CYP4A1, CAR, and PXR followed "Pattern 3". That is the first expression peak appeared at 7–14 days of age, similar to that observed in mice (*Hart et al., 2009*).

## P450-7 family and BA homeostasis

Bile acid (BA) homeostasis is tightly regulated via a feedback loop operated by the nuclear receptors FXR and small heterodimer partner (SHP). Loss of either FXR or SHP alone, or Fxr-/-Shp-/- double knock out mice resulted in cholestasis and liver injury as early as 3 weeks of age, and this dysfunction is linked to the dysregulation of bile acid homeostatic key genes, particularly Cyp7a1 (*Anakk et al., 2011*). Hepatic CYP7A1 is a rate-limiting enzyme that catabolizes cholesterol to bile acids, together with CYP8B1 as the classic BA synthesis pathway, while CYP27A1 and CYP7B1 contribute to the alternative pathway of BA biosynthesis (*Cuesta de Juan et al., 2007*; *Liu et al., 2014*). CYP27A1 is sensitive to inhibition by many xenobiotics (*Lam, Mast & Pikuleva, 2018*). In the present study, the CYPs responsible for cholesterol metabolism and bile acid homeostasis followed "Pattern 2", that was low at the neonatal stage, remained relative high levels throughout the adulthood, and decreased at 800 days of age.

Taken together, CYPs are crucial enzymes in drug metabolism and disposition (*Zanger & Schwab, 2013*). Induction or inhibition of CYPs have been implicated in therapeutic efficacy and toxicity (*Jia et al., 2014*; *Mach et al., 2014*; *Mach et al., 2016*), and age-associated diseases (*Bondarenko et al., 2016*; *Velenosi et al., 2012*) especially in children (*Kilanowicz et al., 2015*; *Li et al., 2017*; *McPhail et al., 2016*) and in elderly (*McPhail et al., 2016*; *Salmin et al., 2017*; *Wauthier, Verbeeck & Calderon, 2007*). Thus, a better understanding of age-associated changes of CYPs is of significance for pharmacology, toxicology, and therapeutics.

## UGT and SULT

Glucuronidation and sulfation are two most important phase-II reactions to conjugate the CYP-catalyzed metabolites for biliary or urinary elimination. Age has significant impact on hepatic activities of glucuronidation and sulfation. For example, porcine hepatic glucuronidation and sulfation activities were low at birth, peaked at 5–10 weeks, and then declined at 20 weeks (*Hu, 2017*), similar to the observations in the present study.

In childhood and adolescence, UGT expression can be affected by hormones and is a reason of individual variation to medication (*Neumann et al., 2016*). Compared to adults, glucuronidation is reduced in children (*Krekels et al., 2012*). UGT1A1 and UGT1A6 are subjected to CAR and PPARα regulation (*Osabe et al., 2008*), and age-associated UGT1A1 changes are paralleled with CYP3A expression alterations in rats fed high-fat diet (*Kawase et al., 2015*; *Osabe et al., 2008*).

Sulfation is the most highly developed pathway during fetal development where glucuronidation in particular is lacking (*Coughtrie, 2015*). Sulfation is normally a detoxification reaction to facilitate the elimination of xenobiotics, although for some

molecules sulfation could be bioactivation (*Coughtrie, 2016*). The decreased Sult1a1 paralleled with major CYP metabolism genes in 600-day old rats (*Mori et al., 2007*), similar to current observations.

## CONCLUSIONS

Overall, the present study characterized age-related changes in a total of 25 CYP isoforms and relevant genes in rat livers from development to aging. In general, these genes are low in neonatal stages, increase with age, but decreased in aged animals, and three expression patterns are characterized. These data could help our better understanding of the effects of CYPs on drug metabolism, pharmacology, and toxicology in the context of maturation and aging.

### Funding
This study is supported by the National Natural Science Foundation of China (81560592, 81560682). The funders had no role in study design, data collection and analysis, decision to publish, or preparation of the manuscript.

### Grant Disclosures
The following grant information was disclosed by the authors:
National Natural Science Foundation of China: 81560592, 81560682.

### Competing Interests
Jie J Liu is an Academic Editor for PeerJ.

### Author Contributions
- Shangfu Xu, Qin Wu and Jie Liu conceived and designed the experiments, performed the experiments, analyzed the data, contributed reagents/materials/analysis tools, prepared figures and/or tables, authored or reviewed drafts of the paper.
- Anling Hu performed the experiments, analyzed the data, prepared figures and/or tables, authored or reviewed drafts of the paper.
- Lu Xie and Jia-Jia Liu performed the experiments.

### Animal Ethics
The following information was supplied relating to ethical approvals (i.e., approving body and any reference numbers):

All animal care and experimental protocols were complied with the Animal Management Guidelines of China and approved by the Animal Use and Care Committee of Zunyi Medical University (2012−02).

### Data Availability
Raw data is available in the Supplemental Files.

## Supplemental Information

Supplemental information for this article can be found online at http://dx.doi.org/10.7717/peerj.7429#supplemental-information.

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
