# Peer review of "Age-associated changes of cytochrome P450 and related phase-2 gene/proteins in livers of rats"

_PeerJ, doi:10.7717/peerj.7429_

## Round 0.1 · original submission · Major Revisions

Please fully address the comments of the reviewers point by point.

Reviewer 1 ·

Basic reporting

No comments

Experimental design

No comments

Validity of the findings

No comments

Additional comments

A research manuscript by Shangfu Xu et al. entitled “Age-associated changes of cytochrome P450 and related phase-2 gene/proteins in livers of rats” submitted to Peer J for critical evaluation. The authors aim to increase our understanding concerning the dynamics of cytochrome P450 and other related genes/proteins in rat liver. The manuscript presents three patterns of gene expression in livers of male rat at fetus, neonates, weaning, puberty, adulthood and aging based on qPCR and western-blot analysis. Even though similar results have been reported in mice, this study focused on the effect of aging on cytochrome P450 in rat still looks relatively novel and might provide some useful information for other researchers to study age-related disorders. However, some concerns should be addressed before publication.
Major concern(s):
This study aims to elucidate the effect of aging on the dynamics of cytochrome P450 expression in rat liver. To eliminate gender-biased results and make the conclusions more convincing, female animals should also be included in this study.
Minor concerns:
1. Line 104, in this part, the results from human mainly focused on neonates and young children, aging-induced changes of these enzymes in human should be added to make it more relevant to this study. Also, please check the mistake of the sentence in line 105.
2. Line 60, please add reference(s) to this sentence.
3. Did not find the supplemental Table 1 in supplemental materials, comments from DMD also missing in supplemental files.
4. It’s a little difficult to follow the results by the figure legends, please show as A:…;B:…;C:…… to briefly introduce each panel.
5. In figure legend of Fig 1, it seems CYP1A1 has been checked but the WB results are CYP1A2. Please confirm this.
6. Why this study did not include the -2d, 1d and 7d samples for the western-blot?

Reviewer 2 ·

Basic reporting

the ms needs to be rewritten since the present form is not acceptable.

Experimental design

This section should be rewritten and I have included my comments in this section

Validity of the findings

Needs more investigations

Additional comments

The study present by Shangfu et al., showed the changes in expression of different CYPs using western blotting and Real time PCR in livers of rats at different ages. The study is interesting and has a novelty that could be added to the field of drug metabolizing enzymes. However, I have major concerns regarding the following points:
1- Introduction: The rationales of the study should be clear and texted in separate paragraph.
2- Authors named CYP families in the introduction as 1, 2, 3, and 4, wheras in Materials and Methods, they named CYPs as CYP4A1; 7A1, 3A1; 2E1 and so on. Authors should clarify the families of CYP and their isozymes in the introduction.
3- Different primers of different CYP genes identified by Real time PCR should be included in Methods.
4- The corresponding enzyme activity for each CYP isozyme should be included to give more evidence to the changes in its corresponding CYP isozyme. For example, dimethylnitrosamine N-demethylase is the corresponding enzyme of CYP2E1. Please see the following citations (PLoS One. 2016 Nov 1;11(11):e0165667; J Helminthol. 2002 Mar;76(1):71-8.; Curr Drug Metab. 2000 Sep;1(2):107-32. Review.)
5- Authors mentioned in page 14 lined 186”The expression of CYP-1 family is shown in Figure 1” and should be corrected into CYP 1A2 since family 1 contains several of CYP isozymes.
6- Authors mentioned in page 14 “Cyp2a1 increased dramatically after birth, reached 250-fold at 200 weanling (21 day), and peaked on 35 days of age (1100 fold). It was gradually declined afterwards.”. This is not true since no change in CYP 2a1 after day 28 until 800 d.
7- Why authors did not study the gene and the protein expression for the same isozyme. They did RTPCR for 1A1 and Western blotting for another isozyme, Cyp2a1. It was very important to perform both expressions for the same CYP since both expressions could confirm each other.
8- My comment No.7 is also the same for figures 2,3 and 4.
9- In page 27, description of CYP 2 family is the same of figure 2 legend. Data of figure 2 should be described in details for each parameter.
10- The same in figure 3.
11- Authors mentioned in Materials and methods that Band intensities were semi-quantified by densitometry using Quantity One® software (version 4.6.2, Bio Rad”. Where are the data of band densities? The band densities of each CYP should be presented as histograms in separate figures.
The manuscript in its present form needs extensive revision.

Reviewer 3 ·

Basic reporting

In my opinon, this manuscript had been written in English clearly, and the structure of this manuscript is good for publishing in PeerJ.

Experimental design

in the experiment design, the major problem is that the enzyme activities of these CYPs in rats should been detected by using their selective probes. Only detecting proteins levels by WB or PCR is not enough to support the results.

Validity of the findings

The data is statistically sound, and the conclusions was appropriately stated, but experiemntal data did not support the conlusion of authors in the present form. The real activities of CYP isoforms should be measured in the revision manuscript to prove the conlusion.

Additional comments

In my opinon, this manuscript had been written in English clearly, and the structure of this manuscript is good for publishing in PeerJ.
But in the experiment design, the major problem is that the enzyme activities of these CYPs in rats should been detected by using their selective probes. Only detecting proteins levels by WB or PCR is not enough to support the results.

---

## Round 0.2 · accepted · Accept

The quality of the reversed version is significantly improved and it is acceptable now.

Reviewer 1 ·

Basic reporting

no comment

Experimental design

no comment

Validity of the findings

no comment

Additional comments

The authors have addressed most of the concerns, I think this revised manuscript is suitable to be published in Peer J.